# Amniotic Fluid Disorders: From Prenatal Management to Neonatal Outcomes

**DOI:** 10.3390/children10030561

**Published:** 2023-03-16

**Authors:** Mor Huri, Mariarosaria Di Tommaso, Viola Seravalli

**Affiliations:** Department of Health Sciences, Division of Obstetrics and Gynecology, University of Florence, 50134 Florence, Italy

**Keywords:** amniotic fluid, amniotic fluid volume, neonatal outcome, oligohydramnios, polyhydramnios, fetal medicine, ultrasound

## Abstract

Amniotic fluid volume assessment has become standard in the surveillance of fetal well-being, especially in high-risk pregnancies. Amniotic fluid disorders are a frequent and important topic in fetal and perinatal medicine. However, although important advances have been achieved, many important and challenging questions remain unanswered to date. An abnormally low amniotic fluid volume, referred to as oligohydramnios, has been traditionally considered a possible indicator of placental insufficiency or fetal compromise and is associated with an increased rate of obstetric interventions. An excess of amniotic fluid, referred to as polyhydramnios, may be secondary to fetal or maternal conditions and has been associated with a variety of adverse pregnancy outcomes, especially when it is severe. The ultrasonographic detection of an amniotic fluid disorder should prompt a proper workup to identify the underlying etiology. Data on the association of isolated oligohydramnios or idiopathic polyhydramnios with adverse obstetric and perinatal outcomes are conflicting. While the management of secondary oligohydramnios is usually guided by the underlying condition, the management of isolated oligohydramnios is poorly defined. Similarly, the management of idiopathic and secondary polyhydramnios is not yet standardized. There is an urgent need for randomized clinical trials to provide stronger recommendations on the management of these two common conditions.

## 1. Introduction

The amniotic fluid (AF) is the fluid that surrounds the fetus in the amniotic cavity during intrauterine development, and is fundamental for proper fetal development and growth in a nonrestricted, sterile, and thermally controlled environment [1,2,3].

Amniotic fluid functions can be categorized as physical, functional and homeostasis [4]. The amniotic fluid protects the fetus from trauma and infection and helps regulate fetal body temperature. It allows fetal movements, and thus the development of the musculoskeletal system, and at the same time prevents compression of the umbilical cord and placenta, protecting the fetus from vascular and nutritional compromise. In addition, fetal swallowing of the amniotic fluid contributes to gastrointestinal tract development [2,5].

The amniotic fluid volume (AFV) is the sum of fluid flowing into and out of the amniotic space. It is the result of a complex interaction of fetal, placental and maternal factors, and appears to be conserved and maintained in a dynamic equilibrium [4,6,7].

In early pregnancy, the AF is isotonic with maternal and fetal plasma, suggesting that the fluid is a transudate either from fetal skin, or from the mother through the uterine decidua or placenta surface. The non-keratinized fetal skin offers no impediment to the movements of fluids, acting as a membrane. Thus, early in gestation, the AF may simply be regarded as an extension of fetal extracellular fluid [7,8,9].

Around 22–25 weeks of gestation, when the fetal skin keratinizes, the AF osmolality and sodium concentration decrease as a result of the production of diluted fetal urine [7]. In late gestation, there are four main volume flows into and out of the amniotic sac. The two major inflows are fetal urine and lung liquid secretion, and the two major outflows are fetal swallowing and intramembranous absorption [10], which consists of the reabsorption of fluid and solutes from the amniotic compartment to the fetal blood via the amnion.

Other routes of production and absorption of AF have been investigated but have been found to contribute little to the AFV. The transmembranous pathway refers to the movement of water and solutes through the surface of the amnion and chorion into the maternal blood within the uterine wall and is extremely small compared to intramembranous absorption [7]. Secretions from fetal oral-nasal cavities into the AF also do not seem to be volumetrically important [10,11].

## 2. Amniotic Fluid Volume Measurement

Amniotic fluid volume evaluation has become standard in the surveillance of fetal well-being, especially in high-risk pregnancies [3,12]. Its assessment is an integral part of ultrasound-based scoring systems, including the Biophysical Profile Score and the modified Biophysical Profile [4,13].

The AFV can be measured directly or indirectly or estimated sonographically. Direct measurement is done at the time of cesarean section or uterine hysterotomy. Indirect measurement is performed via amniocentesis by dye dilution techniques [8]. Despite being the most accurate measures of the AFV, invasive methods are impractical for clinical use. Ultrasound is a non-invasive method and is used widely in clinical practice to evaluate the AFV [5]. Several ultrasound methods are used to assess the AFV. In a subjective assessment, the experience of the operator is essential for reliable results. The most frequently employed semi-quantitative techniques are the measurement of the amniotic fluid index (AFI) and the single deepest vertical pocket (SDVP) [12,14].

The AFI, first proposed by Phelan and Rutherford [15,16], is the summation of the vertical diameter of the largest pocket in each of the four quadrants, using the maternal umbilicus as a central reference point [8]. The SDVP, proposed by Chamberlain [17], is found by selecting the largest vertical measurement with a minimum horizontal measurement of 1 cm. In multiple gestations, the SDVP is the only method used [18].

The advantages of the ultrasound techniques for estimating AFV are that they are simple to perform, easy to teach and reproducible. However, the non-invasive methods have limitations in accuracy related to the fact that ultrasound images are a two-dimensional representation of a complex three-dimensional space. It has been observed that ultrasound assessment is most accurate for identifying normal AFV but is less accurate for identifying an abnormal volume. Magann et al. [19] compared ultrasound assessment with the dye dilution technique and concluded that both semiquantitative techniques are similarly poor predictors and are unreliable for identifying truly abnormal AFV.

Many factors may play a role in the accuracy of ultrasound assessment of the amniotic fluid, such as the experience of the operator, the fetal position, and the possibility of a transient change in the AFV [8,12]. Recently, the use of an automated evaluation of AFI from ultrasound images has been proposed in order to reduce the inter-observer variability of the measurement or the risk of human error [20].

Throughout multiple studies and randomized trials aimed at comparing the use of the AFI with the use of the SDVP in predicting adverse pregnancy outcomes, no single sonographic method has emerged as being superior. The only significant finding has been that the AFI identified more pregnancies as having oligohydramnios than the SDVP. In a systematic review, Nabhan et al. [12] showed that the use of the AFI increases the rate of diagnosis of oligohydramnios compared to the use of the SDVP, leading to superfluous interventions without an improvement in pregnancy outcomes. Many other authors agree that the SDVP measurement is more appropriate for assessing the AFV during fetal surveillance, especially in a population of low-risk pregnancies [5,8,12,21,22,23,24]. Since a higher rate of obstetric intervention can only be justified if there is a demonstrable decrease in adverse outcomes, some scientific societies now recommend the use of the SDVP rather than the AFI to diagnose oligohydramnios in the third trimester [25,26,27,28]. Recently, however, this recommendation was criticized by Wax et al., who suggested that the AFI should be the preferred method for assessing third trimester AFV and diagnosing oligohydramnios [13]. The authors underlined that the primary goal of antepartum surveillance is to reduce the stillbirth rate, and that published studies comparing the two methods are underpowered for evaluating the effect on this important outcome [13].

The SDVP, on the other hand, seems to identify more cases of polyhydramnios compared to the AFI [29]. The low incidence of adverse outcomes found in the presence of an elevated SDVP, but with a normal AFI, suggests that the diagnosis of polyhydramnios should be based on the AFI method [30]. However, further research into optimal methods for the diagnosis of polyhydramnios is needed, and the Society for Maternal Fetal Medicine currently recommends that either method can be used to diagnose polyhydramnios in singleton pregnancies [18].

To improve the accuracy of predicting the risk of adverse outcome, it has also been suggested that ultrasound AFV evaluation should be used in conjunction with other prognostic factors within a prognostic model [3]. This is an interesting approach that deserves to be further investigated.

## 3. Normal Amniotic Fluid Volume

Normal amniotic fluid volume is defined as an AFI between 5 and 25 cm or an SDVP between 2 and 8 cm. Although a single set of thresholds for defining normal AFV is usually used throughout pregnancy for diagnostic purposes, the actual AFV is known to vary each gestational week in a non-linear way. The average AFV increases steadily in early gestation, reaching a peak around 33 weeks. Between 29 and 37 weeks, there is little change in volume, and beyond 39 weeks, the AFV decreases sharply, averaging 515 mL at 41 weeks [7,8,31]. Nomograms for AFV versus gestational age in normal pregnancies have been developed using dye dilution, direct measurement, and ultrasound estimation [8]. The variability in the AFV is proportional to the mean AFV for any given gestational age. Therefore, the AFV varies much less during the first trimester than during the latter half of pregnancy [2].

Near term and in uncomplicated post-term pregnancies, urine concentration increases and the volume of urine production falls, with a relative reduction in the AFV [4]. The cause of this reduction remains unclear, although it may be related to placental involution [32]. It has also been hypothesized that by reducing the AFV at advanced gestation, the fetus may contribute to the initiation of spontaneous uterine contractions by the same mechanism observed in spontaneous or artificial rupture of membranes [1].

Other factors may influence the AFV. Nulliparous women were observed to have significantly lower AFI than parous women in uncomplicated, low-risk pregnancies [33]. It has also been demonstrated that the AFV varies by racial and ethnic group, with differences that seem to be most prominent after 35 weeks and at the extremes of dispersion [34]. Given these physiological factors, the use of universal single thresholds to define abnormal measurement may not be clinically appropriate. Utilization of specific nomograms may enable the application of more stringent gestational age-dependent criteria in detecting abnormal AFV [1]. Nevertheless, only further well-powered studies will demonstrate if the inclusion of these variables in the determination of normal or abnormal AFV leads to improvements in the prediction of adverse pregnancy outcomes [34].

## 4. Oligohydramnios

An AFI less than or equal to 5 cm, or the absence of an SVDP measuring at least 2 cm, is typically used to diagnose oligohydramnios (Figure 1) [12,35]. Anhydramnios is the term used to describe a complete or near-complete lack of amniotic fluid. The prevalence of oligohydramnios varies widely, from approximately 0.5% to 5% of singleton pregnancies, depending on the study population and the diagnostic criteria [32].

Oligohydramnios has been traditionally considered to be a sign of potential adverse perinatal outcome, as well as a possible indicator of placental insufficiency and fetal compromise. Thus, the identification of oligohydramnios usually mandates close fetal surveillance. Furthermore, the diagnosis of oligohydramnios at term is often considered an indication for induction of labor, even in otherwise uncomplicated pregnancies [1,3,36].

Oligohydramnios itself may cause fetal morbidity, especially in the second trimester, when the lack of amniotic fluid may indicate compression of the fetus and induce a series of structural and anatomical deformations (oligohydramnios deformation sequence, or Potter sequence) that are often associated with fetal or neonatal mortality [6]. Furthermore, the presence of severe oligohydramnios during critical periods of gestation may result in fetal pulmonary hypoplasia. Lung hypoplasia may result from the loss of lung fluid: the low intraamniotic pressure increases the alveolar-amniotic gradient and therefore reduces the fluid in the airways, which is the main stimulus for lung development. Moreover, in women with early preterm premature rupture of membranes, oligohydramnios is independently associated with severe neonatal respiratory morbidity and overall mortality [37,38].

During the late second or third trimester, a reduction in AFV may also cause umbilical cord compression, resulting in fetal heart rate decelerations and operative deliveries [6]. Oligohydramnios presents several other challenges: ultrasound visualization is impaired due to lack of contrast and limited fetal mobility, and amniocentesis, when indicated, may be very difficult to perform [4].

With regard to the etiology (Table 1), oligohydramnios may be secondary to fetal anomalies, congenital infections, fetal growth restriction (FGR), post-term pregnancy, premature rupture of membranes, and preeclampsia [1,12,32]. The incidence of major congenital anomalies such as fetal renal agenesis, renal dysplasia, autosomal recessive polycystic kidney disease, and lower urinary tract obstruction is significantly increased in patients with oligohydramnios [6,17,39]. In these patients, initial assessment must include an ultrasound evaluation of fetal anatomy with particular reference to the fetal urinary tract [40]. Oligohydramnios can also result from maternal exposure to medications such as angiotensin-converting enzyme inhibitors, angiotensin receptor blockers or indomethacin; therefore, a history of the use of such medications should be investigated. In monochorionic twin pregnancies, oligohydramnios may manifest as a sign of twin-to-twin transfusion syndrome (TTTS) in the donor twin.

The frequent observation of oligohydramnios in growth-restricted fetuses is related to chronic fetal hypoxemia caused by placental insufficiency, which determines a decreased renal perfusion and fetal urine production rate with the shunting of blood to the fetal heart, adrenals, and brain [3,6,17,21,39,41]. Furthermore, a reduced fetal urine production rate has been suggested to be an early indicator of adverse neonatal outcome [42].

Once oligohydramnios has been detected on ultrasound, a proper workup is needed to investigate the cause (Figure 2) and to decide the correct management, which varies depending on the etiology: e.g., hospitalization and antibiotic therapy in case of preterm premature rupture of membrane after 24 weeks, amniocentesis in case of structural fetal anomalies or for confirmation of suspected congenital infection, medical treatment of infections if available, close fetal surveillance and Doppler studies in case of FGR, and so on.

## 5. Isolated Oligohydramnios

Isolated oligohydramnios refers to the presence of oligohydramnios in an otherwise uncomplicated pregnancy without evidence of fetal structural or chromosomal abnormalities, fetal growth restriction, or infection, and in the absence of maternal hypertensive disorders or renal disease.

In isolated oligohydramnios it remains uncertain why the AFV has decreased. One of the possible pathophysiological mechanisms is an increased absorption of AF due to alterations in the expression of aquaporins, rather than decreased fetal renal blood flow. Indeed, no significant difference was found in the hourly fetal urine production between isolated oligohydramnios and normal AFV [43]. Conversely, a decrease in the fetal urine production rate has been observed in neonates with adverse perinatal outcomes and in growth-restricted fetuses [44]. The renal artery Doppler velocimetry and cerebroplacental ratio were also not found to be significantly different in women with isolated oligohydramnios compared to women with normal AFV [41].

On the other hand, some authors consider isolated oligohydramnios to be a marker of chronic fetal hypoxemia or poor placental function. In a recent study including histopathological analysis of the placenta, a higher rate of placental maternal vascular malperfusion lesions and worse neonatal outcome was demonstrated in pregnancies complicated by isolated term oligohydramnios compared to controls [45]. Based on these findings, the authors suggested that isolated oligohydramnios should be considered to be part of the “placental insufficiency” spectrum, probably in a milder form than preeclampsia and FGR [45]. For these reasons, once- or twice-weekly antenatal fetal surveillance may be considered in patients with isolated oligohydramnios who are not being delivered [28].

While many studies have found that oligohydramnios secondary to maternal or fetal disease is associated with an increased risk of adverse outcome, there are conflicting data on the significance of isolated oligohydramnios and on its association with adverse perinatal outcomes [1,46]. Accordingly, the appropriate management of pregnancy with isolated oligohydramnios at term is still a matter of debate.

In two recent meta-analyses [32,36], isolated oligohydramnios was associated with high obstetric intervention rates (such as induction of labor and cesarean sections). Shrem et al. also described lower Apgar scores and higher rates of admission to neonatal intensive care units in pregnancies with isolated oligohydramnios [36], while Rossi et al. found similar neonatal outcomes compared to pregnancies with normal AFV [32]. Nonetheless, both meta-analyses concluded that there is no clear evidence of the benefits to short- and long-term neonatal outcomes to support induction of labor in cases of isolated oligohydramnios. It is unclear whether the observed associated adverse outcomes reflect the sequelae of the medical intervention or the fetal and maternal comorbidities associated with oligohydramnios, or whether they are a direct result of the reduced AFV itself [32,36].

Nevertheless, a critical outcome that has not yet been fully evaluated is the risk of stillbirth or perinatal mortality in cases of isolated oligohydramnios. In the present literature on isolated oligohydramnios, the number of stillbirths was minimal, in most cases because of a limited sample size, such that most studies did not include stillbirth as an outcome, and those that did so had very few occurrences [46]. Although oligohydramnios in low-risk pregnancies is recognized as an abnormal finding, there is not enough evidence to determine the optimal timing of delivery and therefore to recommend induction of labor with the aim of reducing the risk of adverse outcomes. Only randomized controlled trials comparing different management protocols in cases of isolated oligohydramnios will resolve this issue [36,46]. The only published randomized trial [47] that compared expectant management with induction of labor in cases of isolated oligohydramnios from 40 weeks of gestation was limited by a small sample size and by the lack of blinding, and therefore cannot be used to derive evidence-based conclusions about the best management of this condition.

While the American College of Obstetricians and Gynecologists (ACOG) suggests induction of labor at between 36 and 37 + 6 weeks of gestation in cases of isolated oligohydramnios (or at diagnosis when diagnosed later) [48], Italian guidelines state that there is not enough evidence to support intervention [49]. Other international guidelines on induction of labor do not include specific recommendations on the management of isolated oligohydramnios [50]. Interestingly, a questionnaire survey in the United States revealed that the vast majority of maternal–fetal medicine specialists interviewed consider isolated oligohydramnios an indication for labor induction at 40 weeks, or even at 39 weeks in the presence of a favourable cervix [51].

## 6. Polyhydramnios

Polyhydramnios, or hydramnios, consists of an excess of AF (Figure 3). Polyhydramnios can be diagnosed by either an SDVP ≥ 8 cm, or an AFI ≥ 24 cm or 25 cm (depending on whether the 95th or 97th percentile is used). The prevalence of polyhydramnios is 1–2% of singleton pregnancies [52,53]. The degree of polyhydramnios is frequently categorized as mild, moderate, or severe [18] depending on the value of AFI or SDVP, as shown in Table 2. Polyhydramnios may be further classified as transient, when it resolves spontaneously through gestation, or persistent [54].

Polyhydramnios itself may result in maternal and fetal morbidity. In severe polyhydramnios, the overdistention of the uterus may cause maternal respiratory compromise, and it is believed to increase the risk of preterm delivery, or postpartum hemorrhage due to uterine atony. An increased level of AF can also cause fetal malposition due to excessive fetal mobility, placental abruption from sudden uterine decompression after membrane rupture, or umbilical cord prolapse [4,6]. However, amnioreduction is recommended only as a symptomatic treatment for patients with significant respiratory compromise and discomfort [18,55].

Polyhydramnios may be secondary to either fetal or maternal conditions, or, more rarely, to some placental abnormalities (Table 3). When no etiology can be identified, polyhydramnios is defined as idiopathic. When it is caused by a fetal anomaly, the accumulation of AF can be attributed to three main mechanisms (Table 4): (i) impaired fetal swallowing, caused by fetal abnormalities that compress the fetal trachea, gastrointestinal obstructions such as esophageal atresia, or neuromuscular disorders causing absent or reduced swallowing; (ii) high cardiac output, either caused by severe fetal anemia due to maternal alloimmunization, by some fetal cardiac anomalies, or by fetal or placental tumors, resulting in increased fetal urination; (iii) overproduction of fetal urine due to some renal or urinary tract anomalies. Fetal infections of the TORCH group are also sometimes considered a possible cause of polyhydramnios [4,6,10,18,30], although such association has been questioned [56]. In monochorionic twin pregnancies, polyhydramnios is a sign of TTTS in the recipient fetus.

Maternal gestational or pregestational diabetes is the cause identified in 15 to 24% of cases of polyhydramnios [57,58]. It is hypothesized that maternal hyperglycemia may lead to fetal hyperglycemia with subsequent fetal osmotic diuresis. This hypothesis is supported by the observation that AF glucose concentration often correlates with the AFV [18,59]. The presence of polyhydramnios in pregnancies complicated by maternal diabetes is considered to be a sign of poor glycemic control, and may warrant an adjustment of treatment in addition to closer maternal and fetal monitoring.

Polyhydramnios is also associated with large-for-gestational-age fetuses, and a correlation between neonatal birth weight and AFV has been demonstrated [3,60]. Since larger fetuses have a higher urine output, mild polyhydramnios in these cases can be considered physiologic [18].

Once polyhydramnios has been diagnosed on ultrasound, the workup should focus on identifying the underlying cause (Figure 4). A detailed anatomy scan is indicated to exclude fetal structural anomalies, especially if the polyhydramnios is severe, although sometimes they may not be detectable by prenatal ultrasound. The likelihood of an underlying fetal abnormality increases with the degree of polyhydramnios. Even in the absence of sonographic evidence of fetal anomaly, the likelihood of a major anomaly diagnosed postnatally in the setting of mild, moderate, and severe polyhydramnios is approximately 1%, 2%, and 11%, respectively [18,30,61].

In moderate-to-severe polyhydramnios, when a structural anomaly is present, fetal aneuploidy prevalence is estimated to be around 10%, supporting the role of fetal genetic testing in such cases [30,61]. Moreover, the association of severe polyhydramnios with FGR should raise concern for an underlying genetic syndrome or aneuploidy, such as trisomy 18 or 21 [18,52,62]. When a detailed ultrasound examination demonstrates normal anatomy and growth parameters, the aneuploidy risk is estimated to be 1% or less [61]. Therefore, the presence of polyhydramnios alone is not an absolute indication for genetic testing [52,61]. However, the quality of evidence on this topic is still suboptimal, thereby limiting the drawing of any solid recommendation regarding routine genetic testing in idiopathic polyhydramnios [63].

When polyhydramnios has been diagnosed, screening for gestational diabetes, if not already performed, and assessment of fetal growth should also be recommended [18,64]. Maternal Rh type and the results of a Coombs test should be reviewed, as well as any history of bleeding or trauma during pregnancy, to identify the risk of maternal alloimmunization and fetal anemia [55].

The utility of testing for congenital infections in isolated polyhydramnios has been questioned, especially when polyhydramnios is diagnosed in the third trimester [56,65,66,67]. Congenital infections usually present with additional sonographic findings that may vary depending on the pathogen, such as hyperechogenic bowel, hepato-splenomegaly, placentomegaly, fetal hydrops, or cerebral signs, and the interpretation of maternal serology can be difficult, as results are conclusive only if negative [18,65,66]. Therefore, in the absence of associated ultrasound signs, infectious disease screening is not indicated.

Polyhydramnios has been associated with a variety of adverse pregnancy outcomes, especially if it is severe and/or accompanied by a fetal anomaly [30]. In cases of an identified underlying etiology, the degree of polyhydramnios is associated with an increased rate of preterm delivery, small-for-gestational-age infant, macrosomia, and perinatal mortality [18], and the prognosis seems directly correlated to the underlying cause of polyhydramnios [55]. On the other hand, there is no consensus about the risks associated with idiopathic polyhydramnios. Despite the presence of several studies on polyhydramnios in the literature, many do not stratify by the underlying etiology or the degree of polyhydramnios, and some do not include a control group.

The need for antenatal surveillance in pregnancies complicated by polyhydramnios is supported by some authors, but often without specific recommendations [30]. The ACOG suggests initiating weekly antenatal fetal surveillance at 32 to 34 weeks of gestation for patients with moderate or severe polyhydramnios [28]. Further studies are needed to assess the benefit of antenatal testing and to determine the optimal mode of such testing.

The decision on the timing of delivery is dependent on the severity of polyhydramnios, the association with maternal diabetes, and the presence and type of underlying congenital malformations [55].

## 7. Idiopathic Polyhydramnios

The majority (50–70%) of cases of polyhydramnios are classified as idiopathic [4,18,57,68,69]. Before defining polyhydramnios as idiopathic, all other above-mentioned conditions should be carefully excluded, especially gestational diabetes, which is the most frequent.

The resolution rate of idiopathic polyhydramnios has been reported to be 37%, but it may reach 68% of cases when the degree of polyhydramnios is mild [70,71]. The earlier in pregnancy polyhydramnios is diagnosed, and the milder it is, the more likely it is to resolve during pregnancy [70]. Pregnancy and perinatal outcomes of transient polyhydramnios appear to be similar to those of the general population, with the exception of an increase in neonatal birthweights [54]. On the other hand, polyhydramnios that persists throughout gestation has been associated with an increased risk of adverse maternal and perinatal outcomes compared to transient polyhydramnios and to a normal AFV [54].

There are contrasting results regarding the risk of maternal and neonatal adverse outcomes when polyhydramnios is idiopathic [71]. The majority of cases of idiopathic polyhydramnios are mild, with probable good prognosis and relatively low incidence of associated anomalies. Mild idiopathic polyhydramnios also has a low progression rate to a higher degree of severity [71].

A recent meta-analysis that included twelve studies found an increased risk of stillbirth and neonatal death in pregnancies complicated by idiopathic polyhydramnios compared to pregnancies with normal AFV, as well as greater odds of cesarean deliveries, macrosomia, neonatal acidosis and admission to neonatal intensive care units. Nevertheless, the authors did not stratify the data according to the severity of polyhydramnios because the studies included did not separate mild polyhydramnios from moderate or severe cases, and the authors concluded that further studies are needed, specifically regarding the perinatal risks associated with mild idiopathic polyhydramnios [72].

Some recent studies focused on the outcomes of idiopathic polyhydramnios based on the degree of AFV increase. Vanda et al. [73], in a prospective study on idiopathic polyhydramnios, found an increased risk of postpartum haemorrhage, caesarean section rate and neonatal respiratory distress, and these risks were found to increase with the severity of polyhydramnios. Conversely, Pri-Paz et al. [30], in a retrospective analysis, did not find a significant association between the degree of polyhydramnios and the occurrence of adverse outcomes in the subgroup of cases with idiopathic polyhydramnios. In a retrospective cohort study, Pasquini et al. [71] selected only cases of mild idiopathic polyhydramnios and found that they had an increased risk only of cesarean section compared to controls, in particular, emergency cesarean sections due to abnormal intrapartum cardiotocography or labor dystocia, with similar rates of preterm delivery, postpartum hemorrhage, macrosomia, neonatal acidosis and low Apgar score. Therefore, the authors suggested that mild polyhydramnios should be considered a different condition from moderate and severe polyhydramnios [71]. In particular, patients should be reassured regarding maternal and neonatal outcomes, and the management of their pregnancies should not differ from that of an uncomplicated pregnancy except for the need for increased labor surveillance [71]. None of the above-mentioned studies was powered to evaluate the association of idiopathic polyhydramnios with intrauterine fetal demise.

The controversial data regarding perinatal outcomes associated with idiopathic polyhydramnios are also reflected by the scarce and often conflicting recommendations regarding pregnancy management. While there are studies that have found an association with neonatal morbidity and mortality, the utility of antenatal fetal surveillance for patients with idiopathic polyhydramnios has not been assessed. Although some authors suggest considering antenatal fetal surveillance as part of the management of these pregnancies, at least on an individual basis and after proper counseling and shared decision making between the provider and the patient [30,68,72], currently there is insufficient data about the threshold of the AFV (mild, moderate or severe) at which antenatal testing should be initiated, and further randomized high quality studies are warranted [68,72,74]. We agree with the recommendation of the Society of Maternal Fetal Medicine that antenatal fetal surveillance is not required for the sole indication of mild idiopathic polyhydramnios [18].

The timing of delivery in cases of idiopathic polyhydramnios is also a matter of debate. There is no evidence to indicate that isolated polyhydramnios is associated with poor placental function, and when it is mild it should not be considered an indication for induction of labor [18,71]. Therefore, delivery should be allowed to occur spontaneously at term, and the mode of delivery should be determined based on the usual obstetric indications [18]. The timing of delivery when the degree of polyhydramnios is moderate or severe is still controversial, and since specific recommendations are lacking, the management of such cases depends on local protocols. Women with severe idiopathic polyhydramnios should deliver at a tertiary care center due to the possibility that an undiagnosed underlying anomaly is present [3,18,55].

## 8. Conclusions and Implications for Future Research

Amniotic fluid disorders continue to be a frequent and important topic in fetal and perinatal medicine. Oligohydramnios and polyhydramnios have traditionally been considered indicators of possible adverse perinatal outcomes. Although important advances have been achieved, much is still unknown regarding the physiology, pathophysiology, prognosis and optimal management of these two common conditions.

Many important and challenging questions remain unanswered to date. In this review we have tried to highlight the following:In clinical practice, one set of diagnostic thresholds is used throughout pregnancy to diagnose AFV disorders. Nevertheless, many physiological factors, such as race, parity, and gestational age, may contribute to the actual AFV. Further studies are warranted to clarify whether the inclusion of these variables in the diagnosis improves the detection rate of adverse outcomes. Furthermore, ultrasound semiquantitative methods to measure the amniotic fluid have low accuracy in detecting true abnormal AFV, and it is questionable whether they are necessary in low-risk pregnancies [22].Currently, data on the association of isolated oligohydramnios or idiopathic polyhydramnios with adverse obstetric and perinatal outcomes are conflicting. Further large, well-designed studies should be performed to investigate the association between sonographic AFV disorders and ominous outcomes, in particular intrauterine fetal demise, severe perinatal morbidity and neonatal mortality. For idiopathic polyhydramnios, such studies should be stratified by the degree of polyhydramnios (mild, moderate or severe).Oligohydramnios is associated with an increased rate of obstetric interventions, as it is classically considered an indicator of potential fetal compromise. It is still unclear, however, whether isolated oligohydramnios is an expression of an underlying placental dysfunction and therefore an indication for antepartum fetal surveillance and earlier delivery. Consequently, while the management of secondary oligohydramnios is usually guided by the underlying condition, the best management of isolated oligohydramnios is still poorly defined.Similarly, the management of idiopathic and secondary polyhydramnios has not yet been standardised. A diagnosis of polyhydramnios should prompt identification of the underlying cause. The association with fetal anomalies is stronger with higher severity of polyhydramnios, while mild polyhydramnios should be considered a separate entity. The role of antenatal fetal surveillance and active management in mild, moderate or severe idiopathic polyhydramnios is unclear, especially considering the high rate of cesarean sections associated with all classes of polyhydramnios.There is an urgent need for randomized clinical trials comparing expectant management versus labor induction in the presence of isolated oligohydramnios or moderate to severe idiopathic polyhydramnios, in order to provide stronger recommendations on the management of these conditions.

## Figures and Tables

**Figure 1 children-10-00561-f001:**
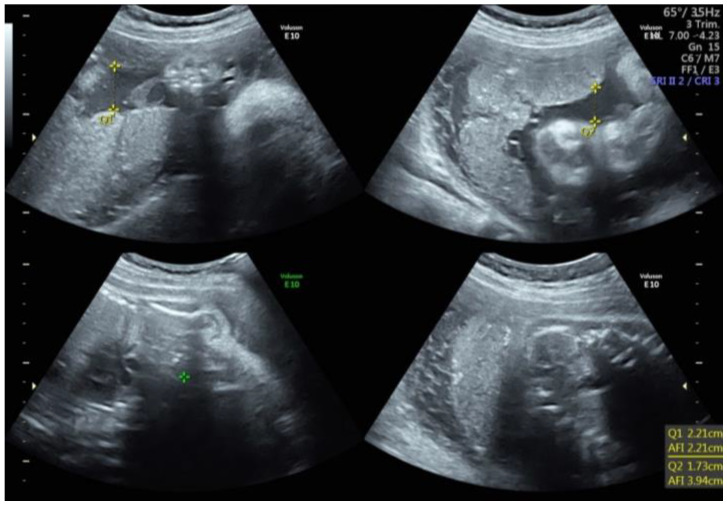
Oligohydramnios. Q: quadrant. AFI: Amniotic Fluid Index.

**Figure 2 children-10-00561-f002:**
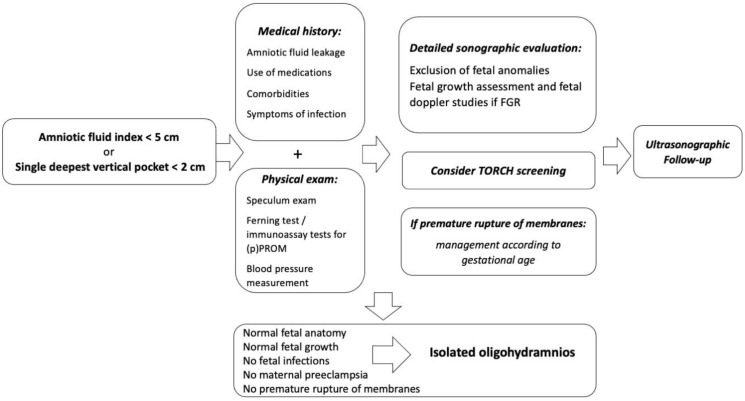
Diagnostic workup for cases of oligohydramnios.

**Figure 3 children-10-00561-f003:**
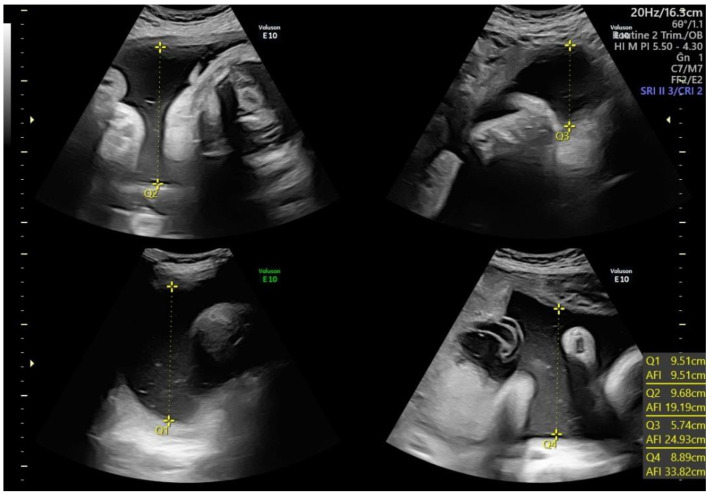
Polyhydramnios. Q: quadrant. AFI: Amniotic Fluid Index.

**Figure 4 children-10-00561-f004:**
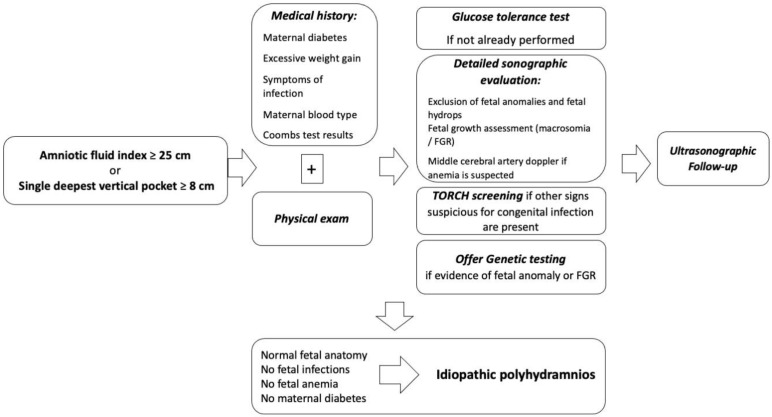
Diagnostic workup for cases of polyhydramnios.

**Table 1 children-10-00561-t001:** Oligohydramnios: etiologies.

Maternal Causes	Fetal Causes	Placental Causes	Isolated Oligohydramnios
Hypertensive disorders Medications (angiotensin-converting enzyme inhibitors, angiotensin receptor blockers, indomethacin); drug abuse	Genitourinary tract abnormalities (lower urinary tract obstruction; Renal anomalies) Congenital infections Stillbirth	Placental insufficiency (fetal growth restriction) Twin-to-twin transfusion syndrome (TTTS) in monochorionic twin pregnancies Post-term pregnancies	No etiology identified
**(Preterm) Premature Rupture of Membranes**			

**Table 2 children-10-00561-t002:** Polyhydramnios: classification by severity.

	Amniotic Fluid Index	Single Vertical Deepest Pocket	Incidence
Mild	25–29.9 cm	8–11 cm	65–70%
Moderate	30–34.9 cm	12–15 cm	20%
Severe	≥35 cm	≥16 cm	<15%

**Table 3 children-10-00561-t003:** Polyhydramnios: etiologies.

Maternal Diabetes (15–24%)	Fetal Causes (11–33%)	Placental Causes	Idiopathic (50–70%)
Gestational diabetes mellitus Pregestational diabetes mellitus (type I; type II)	Fetal malformations Chromosomal anomalies/genetic syndromes Metabolic disorders Fetal anemia Congenital infections	Chorioangioma Twin-to-twin transfusion syndrome (TTTS) in monochorionic twin pregnancies	No etiology identified

**Table 4 children-10-00561-t004:** Fetal anomalies that can cause polyhydramnios, classified by the mechanisms that determine the amniotic fluid increase.

Impaired Swallowing	Increased Cardiac Output/Cardiac Failure	Excessive Urine Production
Intracranial anomalies: - Anencephaly Craniofacial anomalies: - Cleft lip/palate - Micrognathia Intrathoracic masses: - Neck/mediastinal/pulmonary mass (e.g., Cystic adenomatoid malformation of the lung) - Congenital high airway obstruction syndrome - Congenital diaphragmatic hernia Gastrointestinal obstruction/compression: - Congenital diaphragmatic hernia - Esophageal atresia - Duodenal atresia - Other intestinal atresia - Gastroschisis - Omphalocele others Neuromuscular disorders: - Myotonic dystrophy - Fetal akinesia deformation sequence Chromosomal anomalies/genetic syndromes	Sacrococcygeal teratoma Placental chorioangioma Severe Cardiac structural anomalies, e.g.,: - Ebstein anomaly - Tetralogy of Fallot with pulmonary atresia - Cardiomyopathies - Complete atrioventricular block - Supraventricular tachycardia Vascular anomalies Fetal anemia (maternal alloimmunization, parvovirus infection) Fetal thyrotoxicosis	Ureteropelvic junction obstruction Mesoblastic nephroma Bartter syndrome

## Data Availability

Not applicable.

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
