# Peer review of "Amniotic Fluid Disorders: From Prenatal Management to Neonatal Outcomes"

_children, 2023, doi:10.3390/children10030561_

Round 1

Reviewer 1 Report

1. Overall good review of the topic.

2. Suggest one missing area of oligohydramnios morbidity: Tredwell SJ et al. J Prediatr Orthop 2001; 21: 636461; RCT CEMAT Lancet 1998; 351: 242-247

3. Consider Oligo Table with the Mat Fetal Placental like the Poly Table. 

Author Response

Thank you for the opportunity to revise our manuscript. We have now modified the paper in accordance with your suggestions.

We do hope that our revised manuscript is now acceptable for publication.

  1. Overall good review of the topic.

  2. Suggest one missing area of oligohydramnios morbidity: Tredwell SJ et al. J Prediatr Orthop 2001; 21: 636461; RCT CEMAT Lancet 1998; 351: 242-247

    We thank the reviewer for the remark. We have checked the references suggested by the reviewer. However, the first reference refers to tourniquets in Pediatric surgery, while the second reference is the CEMAT trial about the safety of amniocentesis in early versus midtrimester. Regarding the missing area of oligohydramnios morbidity, can the reviewer be more specific?

  3. Consider Oligo Table with the Mat Fetal Placental like the Poly Table. 

    We thank the reviewer for the suggestion. We have now added table 1 (Oligohydramnios: etiologies) as suggested.

Reviewer 2 Report

Line 386 :Therefore, in the absence  of associated ultrasound signs, infectious disease screening is not indicated 

Comment

It is correct that TORCH screening is not indicated as referenced 75 however there are two studies form UK that only CMV is indicated and I have attached the reference below. 

Fayyaz H, Rafi J. TORCH screening in polyhydramnios: an observational study. J Matern Fetal Neonatal Med. 2012 Jul;25(7):1069-72

Abdel-Fattah SA, Bhat A, Illanes S, Bartha JL, Carrington D. TORCH test for fetal medicine indications: only CMV is necessary in the United Kingdom. Prenat Diagn. 2005 Nov;25(11):1028-31

This is a good review and the practice of doing un-necessary TORCH screening and un necessary induction of labour should have been changed.

Overall very well written review however i would suggest that 

Amniotic fluid regulation , measurement , Normal amniotic fluid volume, oligohydramnios . polyhydramnios sections should be brief. Because it feel like more of a BOOK CHAPTER rather than review. Book chapters need full details but i would suggest that Review should not feel like e book chapter because readers loose interest as mainly readers are more interested in clinical aspect in reviews of journals.

Headings 6,7,8 are justified in big details bout not the headings 2,3,4, etc as mentioned.

There is a very fine difference between Journal review and Book Chapter

Author Response

Thank you for the opportunity to revise our manuscript. We have now modified the paper in accordance with your suggestions.

We do hope that our revised manuscript is now acceptable for publication.

1. It is correct that TORCH screening is not indicated as referenced 75 however there are two studies form UK that only CMV is indicated and I have attached the reference below. 

Fayyaz H, Rafi J. TORCH screening in polyhydramnios: an observational study. J Matern Fetal Neonatal Med. 2012 Jul;25(7):1069-72

Abdel-Fattah SA, Bhat A, Illanes S, Bartha JL, Carrington D. TORCH test for fetal medicine indications: only CMV is necessary in the United Kingdom. Prenat Diagn. 2005 Nov;25(11):1028-31

This is a good review and the practice of doing un-necessary TORCH screening and un necessary induction of labour should have been changed. We thank the reviewer for the remarks.

We agree with the reviewer’s comment regarding the TORCH screening. Indeed, Fayyaz et al’s conclusions were similar to our recommendation. Abdel-Fattah et al. in their paper discuss TORCH screening for different fetal ultrasound indications, one of which was polyhydramnios. Although the conclusion is that only CMV screening may be useful in the UK, not one case of polyhydramnios in their cohort was associated with CMV infection, and the most common findings associated with fetal infections in that study were hyperechogenic bowel, ascites, cardiomegaly, and oligohydramnios. We have now included both citations in our manuscript  and reference list.

2. Overall very well written review however i would suggest that Amniotic fluid regulation , measurement , Normal amniotic fluid volume, oligohydramnios . polyhydramnios sections should be brief. Because it feel like more of a BOOK CHAPTER rather than review. Book chapters need full details but i would suggest that Review should not feel like e book chapter because readers loose interest as mainly readers are more interested in clinical aspect in reviews of journals. Headings 6,7,8 are justified in big details bout not the headings 2,3,4, etc as mentioned. There is a very fine difference between Journal review and Book Chapter.

We thank the reviewer for the remark. Following the reviewer’s suggestion, we have now deleted most paragraphs of the section on Amniotic fluid regulation, thus shortening the introduction, and we have deleted the first point of the conclusions on implications for future research. We believe that it is appropriate to keep the sections on Amniotic fluid measurement and Normal amniotic fluid, as they include clinical aspects that may be of interest for the readers, and these sections are also necessary to introduce the following sections on abnormal amniotic fluid volume.